# Associated factors and socio-economic inequality in the prevalence of thinness and stunting among adolescent boys and girls in Uttar Pradesh and Bihar, India

**Pradeep Kumar**[1], **Shobhit Srivastava**[ORCID][1], **Shekhar Chauhan**[2], **Ratna Patel**[3], **Strong P. Marbaniang**[ORCID][3]*, **Preeti Dhillon**[ORCID][1]

**1** Department of Mathematical Demography & Statistics, International Institute for Population Sciences, Mumbai, India, **2** Department of Population Policies and Programmes, International Institute for Population Sciences, Mumbai, India, **3** Department of Public Health and Mortality Studies, International Institute for Population Sciences, Mumbai, India

* marbaniangstrong@gmail.com

**Data Availability Statement:** Data are available from the Harvard Dataverse (https://dataverse.

## Abstract

### Background

Despite economic growth observed in developing countries, under-nutrition still continues to be a major health problem. Undernutrition in adolescence can disrupt normal growth and puberty development and may have long-term impact. Therefore, it is important to study the undernutrition among adolescents. This study aimed to assess the prevalence and the associated factors of stunting, thinness and the coexistence of both (stunting and thinness) among the adolescent belonging to Uttar Pradesh and Bihar, India.

### Methods

The study utilized data from Understanding the Lives of Adolescents and Young Adults (UDAYA) project survey, which was conducted in two Indian states Uttar Pradesh and Bihar, in 2016 by Population Council under the guidance of Ministry of Health and Family Welfare, Government of India. Utilizing information on 20,594 adolescents aged 10–19 years (adolescent boys-5,969 and adolescent girls-14,625), the study examined three outcome variables, i.e., thinness, stunting, and co-existence of both. The study used descriptive and bivariate analysis. Furthermore, the study examined income-related inequality in stunting and thinness through concentration index. At last, the study used Wagstaff decomposition analysis to decompose the concentration index.

### Results

The prevalence of thinness was higher among adolescent boys as compared to girls (25.8 per cent vs. 13.1 per cent). However, stunting was more prevalent among girls (25.6 per cent) than in boys (39.3 per cent). The odds of stunting were higher among late adolescents [Boys- OR:1.79; CI: 1.39, 2.30] and [Girls- OR: 2.25; CI: 1.90,2.67], uneducated

harvard.edu/dataset.xhtml?persistentId=doi:10.
7910/DVN/RRXQNT).

**Funding:** This paper was written using data
collected as part of Population Council's UDAYA
study, which is funded by the Bill and Melinda
Gates Foundation and the David and Lucile Packard
Foundation. No additional funds were received for
the preparation of the paper.

**Competing interests:** The authors have declared
that no competing interests exist.

adolescents [Boys- OR:2.90; CI: 1.67, 5.05] and [Girls- OR: 1.82; CI: 1.44,2.30], and poorest adolescents [Boys- OR:2.54; CI: 1.80, 3.58] and [Girls- OR: 1.79; CI: 1.38,2.32]. Similarly age, educational status, working status and wealth index were significantly associated with thinness among adolescent boys and girls. Media exposure [Boys- OR: 11.8% and Girls- 58.1%] and Wealth index [Boys: 80.1% and Girls: 66.2%] contributed significantly to the inequality in the prevalence of thinness among adolescents. Similarly, wealth index [Boys: 85.2% and Girls: 84.1%] was the only significant contributor to the inequality in the prevalence of stunting among adolescents.

## Conclusion

The study provides an understanding that stunting and thinness is a significant public health concern among adolescents, and there is a need to tackle the issue comprehensively. By tackling the issue comprehensively, we mean that the state government of Uttar Pradesh and Bihar shall screen, assess, and monitor the nutritional status of adolescent boys and girls. The interventions shall focus towards both boys as well as girl adolescents, and particular emphasis should be given to adolescents who belonged to poor households. Also, efforts should be taken by stakeholders to increase family wealth status.

## Introduction

Adolescence is a part of life between childhood and adulthood, from ages 10 to 19 years. It is usually divided into early adolescence (10–14 years) and late adolescence (15–19 years) [1]. It is a crucial stage of human development and an important time for starting the foundation of good health. Today adolescent population constitute about 1.2 billion of the world population (7.2 billion), and nearly 350 million live in South Asia the highest than any other region [2]. In absolute number, India has 253 million adolescent which constitute about 20.9% of the total population [3]. Adolescence is a period of physical growth, cognitive transformation, and reproductive maturation in the life cycle which lead to the high requirements of macro or micronutrients or both [4]. During this period, up to 45% of skeletal growth takes place, and 15% to 25% of adult height is achieved [5]. Nutrition influence growth and development throughout infancy, childhood and adolescent; however, the requirements of nutrient are high during the adolescent period than any other period after birth [6].

Macro and micronutrients deficiency in the body are due to inadequate food intake, low nutrient content of the food and frequent infections. For measuring the nutritional status of adolescent, the World Health Organisation (WHO) has recommended the use of low height-for-age (stunting) and low BMI-for-age (thinness) [7], with the former indicating chronic undernutrition and the later indicating acute undernutrition. Despite economic growth observed in developing countries, under-nutrition still continues to be a major health problem [8]. Recent estimates showed that the prevalence of thinness among Indian adolescent was 26.7% [9], and the prevalence of stunting was 34.1% [1]. Studies from two Indian eastern states (Chhattisgarh and Odisha) shows that the prevalence of thinness among adolescent girls aged 10–14 and 15–19 years was 17.1% and 9.6% respectively [10]. Another study among 5521 adolescent in rural West Bengal shows that the prevalence of stunting was 23.3% among males and 26.9% among females [11]. Existing evidence acknowledge that socioeconomic status, age,

family size, parents education status, lack of latrine, and poor water supply were the commonly mentioned factors that influence the nutritional status of the adolescent [4, 12].

Undernutrition in adolescence can disrupt normal growth and puberty development [6] and increases the risk to infectious diseases [13]. It is also associated with lower educational achievement and income status in adulthood [14]. Undernourished adolescent girls that enter pregnancy are more likely to give birth to a baby of low birth weight or intrauterine growth restricted baby that is more susceptible to metabolic disorder, decrease of growth performance, organ dysfunction and abnormal development, not so good neonatal health, cardiovascular disorder, hormonal imbalance and change in body composition later in life [15]. Nutrition is also vital in reproduction, which includes the safe delivery of infants. MÖLler and Lindmark, (1997) showed that the failure to undergo natural delivery was associated with the height of the mother, which is influence by nutritional status during childhood and adolescent [16].

Given that the adolescent represents the next generation, limited studies have been done to understand the determinants and the inequality of adolescent undernutrition [17–19]. Also, the above-mentioned studies on adolescent nutritional status are based on small sample size and do not represent the scenario of the entire population. Hence, the present study tries to fill this gap using the large sample data from the Comprehensive National Nutritional Survey; this survey provides information about the nutritional status of the adolescent from Uttar Pradesh and Bihar. Given the above background, the present study aimed to assess the prevalence and the associated factors of stunting, thinness and the coexistence of both among the adolescent belonging to Uttar Pradesh and Bihar. The findings will help to provide important evidence for future planning and implementation of nutritional policies and programmes which aimed to improve the nutritional status of adolescents.

## Methods

### Data

The data for this study was carried out from Understanding the Lives of Adolescents and Young Adults (UDAYA) project survey, which was conducted in two Indian states Uttar Pradesh and Bihar, in 2016 by Population Council under the guidance of Ministry of Health and Family Welfare, Government of India. With the written consent of respondents, the survey gathered information on family, media, community environment, assets acquired in adolescence, and quality of transitions to young adulthood indicators. With the use of a multi-stage systematic sampling design, the survey provides the estimates for states as a whole as well as urban and rural areas of the states. The detailed sampling design, data collection procedure, and survey tools available elsewhere [20]. The study treated rural and urban areas of the state as independent sampling domains and, therefore, drew sample areas independently for each of these two domains. The 150 PSUs were further divided equally into rural and urban areas, that is, 75 for rural respondents and 75 for urban respondents. Within each sampling domain, we adopted a multi-stage systematic sampling design. The 2011 census list of villages and wards (each consisting of severalcensus enumeration blocks [CEBs] of 100–200 households) served as the sampling frame for the selection of villages and wards in rural and urban areas, respectively. This list was stratified using four variables, namely, region, village/ward size, proportion of the population belonging to scheduled castes and scheduled tribes, and female literacy. The household sample in rural areas was selected in three stages, while in urban areas it was selected in four stages. In rural areas, villages were first selected systematically from the stratified list as described above, with selection probability proportional to size (PPS). In urban areas, 75 wards were first selected systematically with probability proportional to size, and within each selected ward, CEBs were then arranged by their administrative number and one

CEB was randomly selected. Several CEBs adjacent to the selected CEB were merged to ensure at least 500 households for listing. The study developed a total of five survey tool: a household questionnaire, administered in each selected household; and four individual questionnaires for each of the age groups—younger boys, older boys, younger girls, and older girls, including married girls.

The sample size for Uttar Pradesh and Bihar was 10,350 and 10,350 adolescents aged 10–19 years, respectively. The required sample for each sub-group of adolescents was determined at 920 younger boys, 2,350 older boys, 630 younger girls, 3,750 older girls, and 2,700 married girls in both states. The effective sample size for this study was 20,594 adolescents aged 10–19 years (adolescent boys-5,969 and adolescent girls-14,625) [20]. About 7932 adolescents underwent height and weight measurement. About 7539 adolescents were measure for BMI-for-age Z-score and 7586 Height -for-age Z-score.

The data collection for UDAYA survey was approved by Population Council, New Delhi and ethical review board of Population Council, New Delhi approved the questionnaire that was used in the field work. The written informed consent was elicited from the participants before undertaking the study. This study uses data which is secondary in nature and therefore does not require any ethical approval from any institutional review board.

## Variable description

**Outcome variables.**   There were three outcome variables, i.e., thinness, stunting, and co-existence of both. Thinness among adolescents was constructed using WHO-recommended cut-off points [21]. BMI-for-age Z-score < -2SD was cut-off for thinness. The other variable, i.e., stunting, was constructed using the WHO recommended cut-offs [22]. Height -for-age Z-score of < = -2SD was cut-for stunting among adolescents. The co-existence of thinness and stunting was generated using egen command in STATA 14. It was defined if thinness and stunting co-exist in a single adolescent. The analysis was bifurcated for adolescent boys and adolescent girls because the data only give estimates separately for adolescent boys and girls.

**Explanatory variables.**

1. Age was grouped into two categories i.e., early adolescents (10–14 years) and late adolescents (15–19 years).

2. Education was recorded as no schooling, 1–7, 8–9, and 10 and above years of education.

3. Working status was recoded as not working "no" and working "yes."

4. Media exposure was coded as no exposure, rare exposure, and frequent exposure.

5. Wealth index was recoded as poorest, poorer, middle, richer, and richest. The survey measured household economic status, using a wealth index composed of household asset data on ownership of selected durable goods, including means of transportation, as well as data on access to a number of amenities. The wealth index was constructed by allocating the following scores to a households reported assets or amenities. Than using the scores were divided into five quintiles.

6. Caste was recoded as Scheduled caste and Scheduled tribe (SC/ST) and non-SC/ST.

7. Religion was recoded as Hindu and non-Hindu. The category of non-Hindu was recoded as so because the frequency of other religions was very low; therefore, analytical purpose the recoding was done in a respective manner.

8. The residence was available in data as urban and rural.

9. Data were available for two states i.e., Uttar Pradesh and Bihar, as the survey was conducted in these two states only.

## Statistical analysis

Descriptive and bivariate analysis was done to understand the sample distribution of the study population and get the estimates of thinness, stunting, and co-existence of both separately for adolescent boys and girls. Further, multivariable logistic regression analysis was used to identify the factors associated with thinness, stunting, and co-existence of both. Lastly, regression-based decomposition analysis was used to find the absolute contribution of the factors for outcome variables.

## Concentration Index (CI)

Income-related inequality in infant mortality was quantified by the concentration index (CI) and the concentration curve (CC), using the wealth score as the socio-economic indicator and binary outcome as thinness, stunting, and co-existence of both. The concentration curve is obtained by plotting the cumulative proportion of poor health against the cumulative proportion of the population ranked by the socio-economic indicator. The concentration index can be written as follows:

$$C = \frac{2}{\mu} cov(y_i, R_i)$$

Where, C is the concentration index; $y_i$ is the outcome variable index; $R$ is the fractional rank of individual $i$ in the distribution of socio-economic position; $\mu$ is the mean of the outcome variable of the sample and $cov$ denotes the covariance [23].

The index value lies between -1 to +1. If the curve lies above the line of equality, the concentration index takes a negative value, indicating a disproportionate concentration of inequality among the poor (pro-rich). Conversely, if the curve lies below the line of equality, the concentration index takes a positive value, indicating a disproportional concentration of inequality among the rich (pro-poor). In the absence of socio-economic related inequality, the concentration index is zero. In simpler terms, for this paper, pro-rich inequality means that the stunting and thinness is highly prevalent among the adolescents belonging to poor household and vice-versa.

## Decomposition of the concentration index

The study used Wagstaff decomposition analysis to decompose the concentration index. Wagstaff's decomposition demonstrated that the concentration index could be decomposed into the contributions of each factor to the income-related inequalities [24]. Based on the linear regression relationship between the outcome variable $y_i$, the intercept α, the relative contribution of $x_{ki}$ and the residual error $\varepsilon_i$

$$y_i = \alpha + \sum \beta_k x_{ki} + \varepsilon_i$$

Where $\varepsilon_i$ is an error term, given the relationship between $y_i$ and $x_{ki}$, the CI for y (C) can be rewritten as:

$$C = \sum \left(\frac{\beta_x \bar{x}_k}{\mu}\right) C_k + \frac{GC\varepsilon}{\mu}/\mu$$

Where $\mu$ is the mean of $y_i$, $\bar{x}_k$, is the mean of $x_k$, $\beta_k$ is the coefficient from a linear regression of

outcome variable, $C_k$ is the concentration index for $x_k$ (defined analogously to C, and $GC_\varepsilon$ is the generalized concentration index for the error term ($\varepsilon_i$).

Here C is the outcome of two components: First, the determinants or 'explained' factors. The explained factors indicate that the proportion of inequalities in the outcome (thinness, stunting, and co-existence of both) variable is explained by the selected various socio-economic groups, i.e., $x_k$. Second, a residual or 'unexplained' factor $\left(\frac{GC_\varepsilon}{\mu}/\mu\right)$, indicating the inequality in health variables that cannot be explained by selected explanatory factors across various socio-economic groups.

## Results

Fig 1 display the prevalence of thinness, stunting, and co-existing of both among adolescents aged 10–19 years. The prevalence of thinness (25.8 per cent vs. 13.1 per cent) was higher among adolescent boys as compared to girl counterparts. However, the stunting was more prevalent among girls (25.6 per cent) than in boys (39.3 per cent). Nearly 10 per cent of adolescent boys and six per cent of adolescent girls suffered from co-existence of both indicators.

The socio-demographic profile of the study population was presented in Table 1. A higher proportion of the study population was from the late adolescents' group (boys-65 per cent and girls-88.7 per cent), and about one-fourth of boy and one-third of adolescent girls had 10 & above years of schooling. Nearly 27 per cent of adolescent boys and 17 per cent adolescent girls were working, and three-fourth of boys and about half of the adolescent girls used mass media frequently.

Table 2 shows the percentage distribution of thinness, stunting, and co-existence of both among adolescents aged 10–19 years by various background characteristics. The prevalence of thinness was higher among early adolescents groups, irrespective of their gender. Moreover, stunting was more prevalent among late adolescents (boy-27.3 per cent and girl-42.9 per cent). Nearly 11 per cent of boys and 8 per cent of early adolescent girls suffered from co-existence of both. The prevalence of thinness (30.4 per cent) and co-existence of both (13 per cent) were highest among adolescent boys who had 1–7 years of schooling, and it was least among those

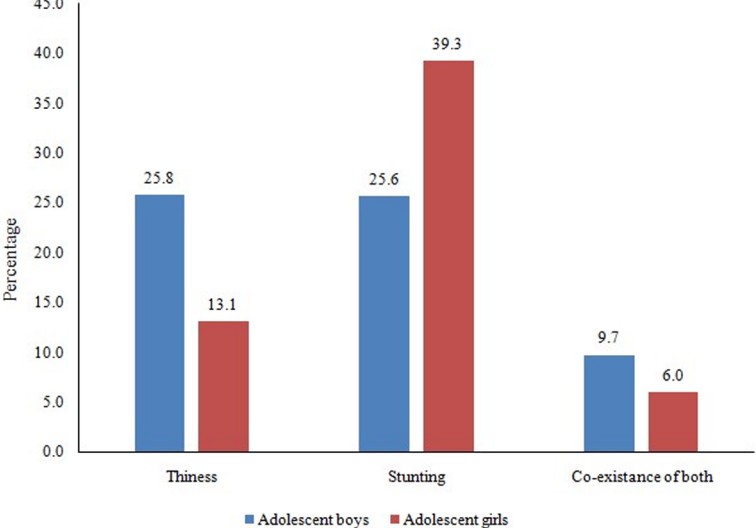

**Fig 1. Thinness, stunting, and co-existence of both among adolescents aged 10–19 years.**

**Table 1. Socio-demographic profile of adolescents aged 10–19 years.**

| Background characteristics | Adolescent boys | | Adolescent girls | |
|---|---|---|---|---|
| | Sample | Percentage | Sample | Percentage |
| **Age (years)** | | | | |
| Early adolescents (10–14) | 2,084 | 34.9 | 1,653 | 11.3 |
| Late adolescents (15–19) | 3,885 | 65.1 | 12,972 | 88.7 |
| **Educational status (years)** | | | | |
| No schooling | 190 | 3.2 | 1,890 | 12.9 |
| 1–7 | 2,497 | 41.8 | 3,939 | 26.9 |
| 8–9 | 1,754 | 29.4 | 4,093 | 28.0 |
| 10 and above | 1,528 | 25.6 | 4,703 | 32.2 |
| **Working status** | | | | |
| No | 4,377 | 73.3 | 12,179 | 83.3 |
| Yes | 1,592 | 26.7 | 2,446 | 16.7 |
| **Media exposure** | | | | |
| No exposure | 335 | 5.6 | 2,703 | 18.5 |
| Rare | 1,078 | 18.1 | 4,212 | 28.8 |
| Frequent | 4,555 | 76.3 | 7,710 | 52.7 |
| **Wealth Index** | | | | |
| Poorest | 704 | 11.8 | 1,971 | 13.5 |
| Poorer | 1,193 | 20.0 | 2,735 | 18.7 |
| Middle | 1,374 | 23.0 | 3,188 | 21.8 |
| Richer | 1,391 | 23.3 | 3,577 | 24.5 |
| Richest | 1,308 | 21.9 | 3,154 | 21.6 |
| **Caste** | | | | |
| SC/ST | 1,605 | 26.9 | 3,784 | 25.9 |
| Non-SC/ST | 4,364 | 73.1 | 10,841 | 74.1 |
| **Religion** | | | | |
| Hindu | 5,024 | 84.2 | 11,540 | 78.9 |
| Non-Hindu | 945 | 15.8 | 3,085 | 21.1 |
| **Residence** | | | | |
| Urban | 1,030 | 17.3 | 2,356 | 16.1 |
| Rural | 4,939 | 82.7 | 12,269 | 83.9 |
| **States** | | | | |
| Uttar Pradesh | 4,069 | 68.2 | 9,855 | 67.4 |
| Bihar | 1,900 | 31.8 | 4,770 | 32.6 |
| **Total** | 5,969 | 100.0 | 14,625 | 100.0 |

SC/ST: Scheduled Caste/Scheduled Tribe.

who had 10 & above years of education. In the case of adolescent girls, a negative association was found between years of schooling and co-existence of thinness and stunting. Thinness was higher among adolescents who were not working (boy-27.6 per cent and girl-13.5 per cent), and this finding was revered for stunting. The prevalence of thinness (boy-23.2 per cent and girl-12.1 per cent), stunting (boy-23.9 per cent and girl-36.8 per cent), and co-existence of both (boy-8 per cent and girl-4.8 per cent) were lowest among adolescents who used mass media frequently.

On the other hand, thinness, stunting, and co-existence of both have a negative relationship with wealth index for adolescent boys. For instance, these indicators was highest among

**Table 2. Percentage distribution for thinness, stunting and co-existence of both by background characteristics among adolescents aged 10–19 years.**

| Background characteristics | Adolescent boys | | | Adolescent girls | | |
|---|---|---|---|---|---|---|
| | Thinness (N = 3242) | Stunting (N = 3190) | Both (N = 3185) | Thinness (N = 4297) | Stunting (N = 4396) | Both (N = 3979) |
| **Age (years)** | | | | | | |
| Early adolescents (10–14) | 29.3 | 24.5 | 11.3 | 19.0 | 32.3 | 7.6 |
| Late adolescents (15–19) | 20.7 | 27.3 | 7.3 | 9.7 | 42.9 | 5.0 |
| **Educational status (years)** | | | | | | |
| No schooling | 23.2 | 29.0 | 6.8 | 12.7 | 49.6 | 7.4 |
| 1–7 | 30.4 | 28.1 | 13.0 | 16.7 | 38.5 | 7.1 |
| 8–9 | 21.9 | 24.3 | 6.9 | 9.6 | 38.2 | 5.3 |
| 10 and above | 14.8 | 17.3 | 1.9 | 10.3 | 36.0 | 3.9 |
| **Working status** | | | | | | |
| No | 27.6 | 24.5 | 10.6 | 13.5 | 38.9 | 6.0 |
| Yes | 19.3 | 30.0 | 6.6 | 10.1 | 42.2 | 6.0 |
| **Media exposure** | | | | | | |
| No exposure | 32.2 | 28.3 | 11.3 | 14.9 | 41.4 | 8.5 |
| Rare | 33.7 | 31.4 | 15.7 | 13.6 | 42.3 | 6.5 |
| Frequent | 23.2 | 23.9 | 8.0 | 12.1 | 36.8 | 4.8 |
| **Wealth Index** | | | | | | |
| Poorest | 33.9 | 35.2 | 16.5 | 17.2 | 44.9 | 9.3 |
| Poorer | 32.3 | 30.3 | 12.0 | 11.7 | 45.3 | 6.5 |
| Middle | 29.1 | 29.5 | 12.1 | 13.3 | 42.1 | 7.7 |
| Richer | 20.6 | 22.6 | 6.5 | 12.2 | 37.9 | 4.3 |
| Richest | 16.5 | 14.4 | 4.0 | 12.0 | 28.2 | 3.3 |
| **Caste** | | | | | | |
| SC/ST | 31.0 | 33.4 | 11.8 | 13.7 | 45.6 | 8.0 |
| Non-SC/ST | 23.9 | 22.8 | 9.0 | 12.8 | 36.8 | 5.2 |
| **Religion** | | | | | | |
| Hindu | 25.9 | 26.2 | 10.1 | 12.3 | 40.1 | 5.9 |
| Non-Hindu | 25.6 | 22.5 | 7.9 | 16.1 | 36.1 | 6.1 |
| **Residence** | | | | | | |
| Urban | 20.4 | 19.0 | 5.5 | 12.5 | 36.0 | 4.2 |
| Rural | 26.9 | 27.0 | 10.6 | 13.2 | 39.8 | 6.3 |
| **States** | | | | | | |
| Uttar Pradesh | 25.5 | 27.3 | 10.0 | 14.1 | 38.0 | 6.1 |
| Bihar | 26.4 | 22.3 | 9.1 | 11.1 | 42.1 | 5.6 |

SC/ST: Scheduled Caste/Scheduled Tribe.

poorest adolescents (thinness-33.9 per cent, stunting-35 per cent and co-existence-16.5 per cent) and lowest in richest ones (thinness-16.5 per cent, stunting-14.4 per cent and co-existence-4 per cent). In the case of adolescent girls, a similar relationship was found for the co-existence of both indicators. The prevalence of thinness (boy-31 per cent and girl-13.7 per cent), stunting (boy-33.4 per cent and girl-45.6 per cent), and co-existence of both (boy-11.8 per cent and girl-8 per cent) were highest among SC/ST adolescents compared to non-SC/ST ones, irrespective to their gender. Similarly, rural adolescents suffered more from thinness, stunting, and co-existence of both than urban counterparts, irrespective of their gender.

Z-scores for thinness and stunting by background characteristics is given in Table 3. The z-score of BMI for age were highest among early adolescents (boys: -1.43 and girls: -1.10), andz-

**Table 3. Z-score for thinness and stunting by background characteristics among adolescents aged 10–19 years.**

| Background characteristics | Adolescent boys | | Adolescent girls | |
|---|---|---|---|---|
| | Z-score BMI for age (N = 3242) | Z-score Height for age (N = 3190) | Z-score BMI for age (N = 4297) | Z-score Height for age (N = 4396) |
| **Age (years)** | | | | |
| Early adolescents (10–14) | -1.43 | -1.16 | -1.10 | -1.49 |
| Late adolescents (15–19) | -1.26 | -1.49 | -0.70 | -1.84 |
| **Educational status** | | | | |
| No schooling | -1.40 | -1.41 | -0.75 | -1.96 |
| 1–7 | -1.49 | -1.27 | -0.98 | -1.64 |
| 8–9 | -1.26 | -1.34 | -0.73 | -1.75 |
| 10 and above | -0.99 | -1.28 | -0.71 | -1.70 |
| **Working status** | | | | |
| No | -1.37 | -1.24 | -0.82 | -1.70 |
| Yes | -1.31 | -1.50 | -0.90 | -1.85 |
| **Media exposure** | | | | |
| No exposure | -1.46 | -1.44 | -0.88 | -1.83 |
| Rare | -1.54 | -1.38 | -0.83 | -1.87 |
| Frequent | -1.30 | -1.26 | -0.81 | -1.60 |
| **Wealth index** | | | | |
| Poorest | -1.58 | -1.52 | -0.98 | -1.96 |
| Poorer | -1.56 | -1.48 | -0.94 | -1.88 |
| Middle | -1.52 | -1.44 | -0.83 | -1.83 |
| Richer | -1.28 | -1.19 | -0.79 | -1.65 |
| Richest | -0.95 | -0.92 | -0.66 | -1.36 |
| **Caste** | | | | |
| SC/ST | -1.52 | -1.54 | -0.84 | -1.91 |
| Non-SC/ST | -1.30 | -1.20 | -0.82 | -1.65 |
| **Religion** | | | | |
| Hindu | -1.36 | -1.30 | -0.82 | -1.75 |
| Non-Hindu | -1.36 | -1.25 | -0.84 | -1.61 |
| **Residence** | | | | |
| Urban | -1.03 | -1.00 | -0.72 | -1.60 |
| Rural | -1.43 | -1.35 | -0.85 | -1.74 |
| **States** | | | | |
| Uttar Pradesh | -1.37 | -1.35 | -0.83 | -1.69 |
| Bihar | -1.33 | -1.18 | -0.82 | -1.79 |
| **Total** | -1.36 | -1.29 | -0.83 | -1.72 |

SC/ST: Scheduled Caste/Scheduled Tribe.

score of height for age were more among late adolescents (boys: -1.49 and girls: -1.84). More-over, there was a positive association between wealth index and, z-score of BMI for age and z-score of height for age. Adolescents who belonged to SC/ST group had higher z-score of BMI for age and height for age than non-SC/ST caste group.

Estimates from logistic regression analysis for thinness, stunting, and co-existence of both among adolescents aged 10–19 years were presented in Table 4. Stunting was 79 per cent sig-nificantly more [OR: 1.79; CI: 1.39–2.3] in late adolescent boys than early ones. In case of ado-lescent girls, thinness was 52 per cent less [OR: 0.48; CI: 0.37–0.62], stunting was 2.25 times

**Table 4. Logistic regression estimates for thinness, stunting and co-existence of both by background characteristics among adolescents aged 10–19 years.**

| Background characteristics | Adolescent boys | | | Adolescent girls | | |
|---|---|---|---|---|---|---|
| | Thinness (N = 3242) | Stunting (N = 3190) | Both (N = 3185) | Thinness (N = 4297) | Stunting (N = 4396) | Both (N = 3979) |
| | OR (95% CI) | OR (95% CI) | OR (95% CI) | OR (95% CI) | OR (95% CI) | OR (95% CI) |
| **Age (years)** | | | | | | |
| Early adolescents (10–14) | Ref. | Ref. | Ref. | Ref. | Ref. | Ref. |
| Late adolescents (15–19) | 1.14(0.89,1.45) | 1.79*(1.39,2.30) | 1.35(0.91,1.99) | 0.48*(0.37,0.62) | 2.25*(1.90,2.67) | 0.62*(0.43,0.89) |
| **Educational status (years)** | | | | | | |
| No schooling | 2.02*(1.15,3.54) | 2.90*(1.67,5.05) | 5.47*(2.07,14.5) | 1.02(0.7,1.48) | 1.82*(1.44,2.30) | 1.62(0.91,2.89) |
| 1–7 | 1.93*(1.39,2.7) | 2.13*(1.52,2.99) | 4.83*(2.38,9.8) | 0.91(0.65,1.27) | 1.54*(1.25,1.90) | 1.29(0.76,2.21) |
| 8–9 | 1.45*(1.07,1.97) | 1.43*(1.05,1.96) | 2.65*(1.32,5.33) | 0.61*(0.44,0.86) | 1.18(0.97,1.43) | 0.87(0.5,1.49) |
| 10 and above | Ref. | Ref. | Ref. | Ref. | Ref. | Ref. |
| **Working status** | | | | | | |
| No | 1.38*(1.07,1.76) | 0.95(0.75,1.21) | 1.72*(1.13,2.62) | 1.25(0.91,1.73) | 0.89(0.73,1.09) | 1.04(0.67,1.62) |
| Yes | Ref. | Ref. | Ref. | Ref. | Ref. | Ref. |
| **Media exposure** | | | | | | |
| No exposure | Ref. | Ref. | Ref. | Ref. | Ref. | Ref. |
| Rare | 1.25(0.82,1.9) | 1.15(0.75,1.77) | 1.42(0.77,2.59) | 1.11(0.82,1.51) | 1.12(0.92,1.37) | 0.91(0.59,1.39) |
| Frequent | 1.01(0.68,1.5) | 1.01(0.67,1.51) | 0.99(0.55,1.78) | 0.95(0.69,1.3) | 1.02(0.83,1.25) | 0.81(0.52,1.26) |
| **Wealth index** | | | | | | |
| Poorest | 1.69*(1.22,2.35) | 2.54*(1.8,3.58) | 3.11*(1.82,5.31) | 1.27(0.87,1.87) | 1.79*(1.38,2.32) | 1.84*(1.02,3.33) |
| Poorer | 1.39*(1.03,1.88) | 2.0*(1.46,2.75) | 2.19*(1.3,3.66) | 1.06(0.74,1.51) | 1.70*(1.34,2.15) | 1.57(0.89,2.76) |
| Middle | 1.48*(1.14,1.93) | 2.0*(1.5,2.65) | 2.23*(1.38,3.6) | 1.22(0.89,1.68) | 1.59*(1.29,1.97) | 2.17*(1.32,3.57) |
| Richer | 1.35*(1.06,1.73) | 1.48*(1.12,1.95) | 1.70*(1.05,2.74) | 1.13(0.85,1.51) | 1.40*(1.15,1.69) | 1.33(0.81,2.16) |
| Richest | Ref. | Ref. | Ref. | Ref. | Ref. | Ref. |
| **Caste** | | | | | | |
| SC/ST | 1.26*(1.03,1.53) | 1.29*(1.05,1.59) | 1.12(0.82,1.52) | 1.13(0.89,1.42) | 1.31*(1.13,1.53) | 1.34(0.96,1.89) |
| Non-SC/ST | Ref. | Ref. | Ref. | Ref. | Ref. | Ref. |
| **Religion** | | | | | | |
| Hindu | Ref. | Ref. | Ref. | Ref. | Ref. | Ref. |
| Non-Hindu | 1.07(0.85,1.35) | 0.98(0.76,1.27) | 0.87(0.58,1.3) | 1.31*(1.03,1.66) | 0.93(0.78,1.1) | 1.35(0.94,1.93) |
| **Residence** | | | | | | |
| Urban | Ref. | Ref. | Ref. | Ref. | Ref. | Ref. |
| Rural | 1.13(0.93,1.36) | 1.15(0.94,1.41) | 1.31(0.95,1.81) | 1.0(0.81,1.25) | 0.97(0.84,1.12) | 0.94(0.68,1.31) |
| **States** | | | | | | |
| Uttar Pradesh | Ref. | Ref. | Ref. | Ref. | Ref. | Ref. |
| Bihar | 0.82*(0.69,0.97) | 0.66*(0.55,0.79) | 0.61*(0.46,0.8) | 0.74*(0.61,0.9) | 0.97(0.85,1.1) | 0.71*(0.52,0.95) |

Ref: Reference; OR: Odds Ratio;

* if p<0.05; CI: Confidence Interval; % Percentage.

more [OR: 2.25; CI: 1.9–2.67] and presence of both was 38 per cent less [OR: 0.62; CI: 0.43–0.89] in late adolescents compared to early ones. The odds of thinness [OR: 2.02; CI: 1.15–3.54], stunting [OR: 2.9; CI: 1.67–5.05], and presence of both [OR: 5.47; CI: 2.07–14.5] was 2.02 times,1.15 times cent and 2.9 times significantly more among adolescent boys who had no schooling respectively than those had 10 and above schooling. Moreover, the likelihood of thinness [OR: 1.38; CI: 1.07–1.76] and co-existence of both (thinness and stunting) [OR: 1.72; CI: 1.13–2.62] was 38 per cent and 72 per cent significantly more likely among not working adolescent boys respectively than working counterparts. With reference to richest category,

adolescent boys who were belonged to poorest families, 1.69 times, 2.54 times, and 3.11 times more to suffer from thinness [OR: 1.69; CI: 1.22–2.35], stunting [OR: 2.54; CI: 1.8–3.58] and co-existence of both [OR: 3.11; CI: 1.82–5.31], respectively. Similarly, stunting [OR: 1.79; CI: 1.38–2.32], and co-existence of both (thinness and stunting) [OR: 1.84; CI: 1.02–3.33] were 79 per cent and 84 per cent significantly more in adolescent girls who belonged to poorest wealth quintile. Moreover, SC/ST adolescent boys were 26 per cent and 29 per cent significantly more to suffer from thinness [OR: 1.26; CI: 1.03–1.53] and stunting [OR: 1.29; CI: 1.05–1.59], respectively compared to non-SC/ST counterparts. The odds of stunting were 31 per cent significantly more among adolescent girls [OR: 1.31; CI: 1.33–1.53] who belonged to the SC/ST community than non-SC/ST ones.

Fig 2A–2C presents the concentration curve for thinness, stunting and co-existence of both among adolescent boys and girls aged 10–19 years. If the curve is formed below the line of equality than the inequality is concentrated towards rich and vice-versa. Moreover, more the

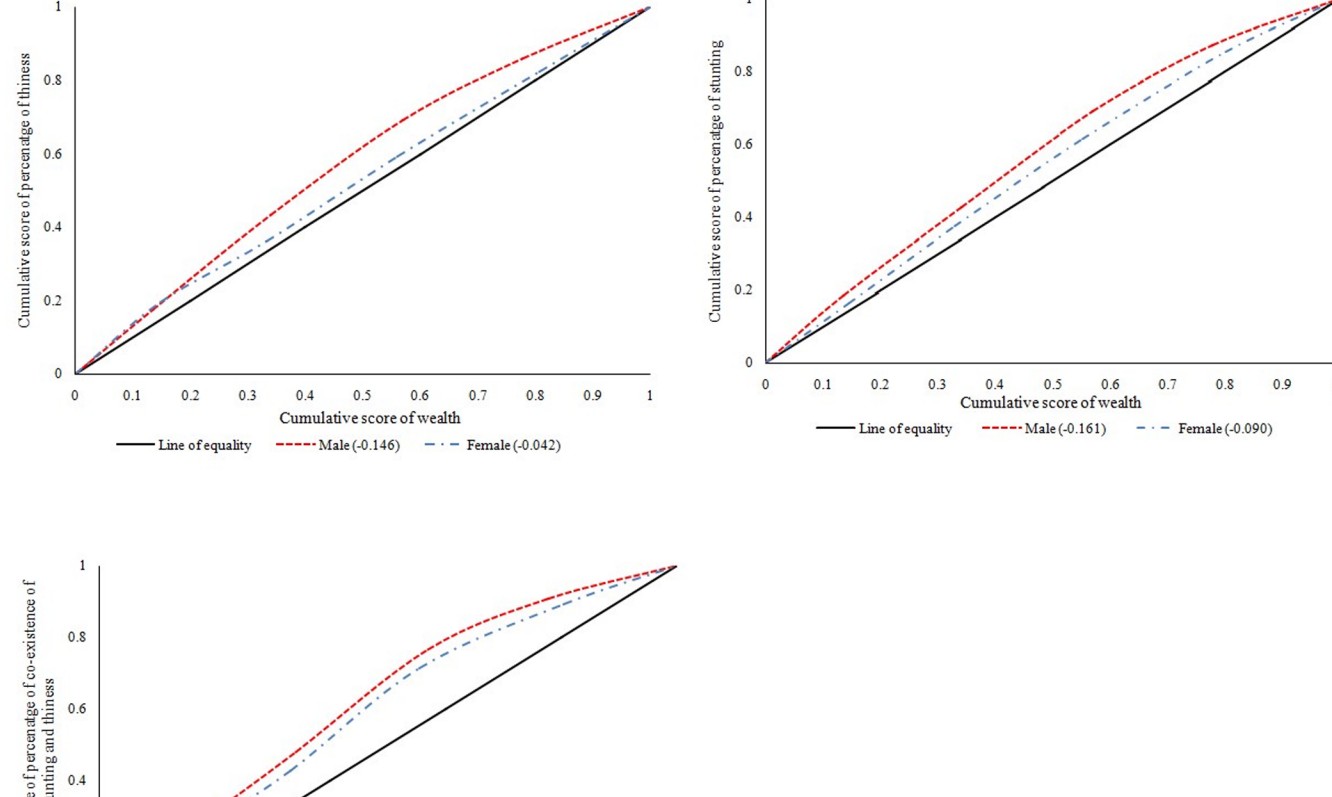

**Fig 2. a** Concentration curve for thinness among adolescents aged 10–19 years. *Male refers to adolescent boys and Female refers to adolescent girls.* **b** Concentration curve for stunting among adolescents aged 10–19 years. *Male refers to adolescent boys and Female refers to adolescent girls.* **c** Concentration curve for co-existence of stunting and thinness among adolescents aged 10–19 years. *Male refers to adolescent boys and Female refers to adolescent girls.*

**Table 5. Estimates of decomposition analysis for contribution of various explanatory variables for thinness among adolescents aged 10–19 years.**

| Background characteristics | Adolescent boys | | | | | Adolescent girls | | | | |
|---|---|---|---|---|---|---|---|---|---|---|
| | Elasticity | CI | Absolute contribution | % contribution | Total | Elasticity | CI | Absolute contribution | % contribution | Total |
| **Age (years)** | | | | | | | | | | |
| Early adolescents (10–14) | | | | | | | | | | |
| Late adolescents (15–19) | 0.006 | 0.087 | 0.001 | -1.3 | -1.3 | -0.060 | 0.016 | -0.001 | 16.1 | 16.1 |
| **Educational status (years)** | | | | | | | | | | |
| No schooling | | | | | | | | | | |
| 1–7 | 0.032 | -0.092 | -0.003 | 8.1 | | -0.002 | -0.088 | 0.000 | -3.2 | |
| 8–9 | -0.001 | 0.064 | 0.000 | 0.3 | | -0.007 | 0.031 | 0.000 | 3.2 | |
| 10 and above | -0.009 | 0.268 | -0.002 | 6.5 | 14.8 | 0.001 | 0.277 | 0.000 | -4.8 | -4.8 |
| **Working status** | | | | | | | | | | |
| No | | | | | | | | | | |
| Yes | -0.022 | -0.152 | 0.003 | -9.1 | -9.1 | -0.004 | -0.255 | 0.001 | -17.7 | -17.7 |
| **Media exposure** | | | | | | | | | | |
| No exposure | | | | | | | | | | |
| Rare | 0.006 | -0.304 | -0.002 | 5.1 | | -0.003 | -0.226 | 0.001 | -9.7 | |
| Frequent | -0.021 | 0.120 | -0.003 | 6.7 | 11.8 | -0.016 | 0.261 | -0.004 | 67.7 | 58.1 |
| **Wealth Index** | | | | | | | | | | |
| Poorest | | | | | | | | | | |
| Poorer | 0.000 | -0.526 | 0.000 | -0.5 | | -0.010 | -0.518 | 0.005 | -82.3 | |
| Middle | -0.006 | -0.091 | 0.001 | -1.3 | | -0.006 | -0.107 | 0.001 | -11.3 | |
| Richer | -0.021 | 0.346 | -0.007 | 19.6 | | -0.010 | 0.367 | -0.004 | 59.7 | |
| Richest | -0.030 | 0.778 | -0.023 | 62.4 | 80.1 | -0.008 | 0.808 | -0.006 | 100.0 | 66.2 |
| **Caste** | | | | | | | | | | |
| SC/ST | | | | | | | | | | |
| Non-SC/ST | -0.043 | 0.089 | -0.004 | 10.2 | 10.2 | -0.012 | 0.086 | -0.001 | 16.1 | 16.1 |
| **Religion** | | | | | | | | | | |
| Hindu | | | | | | | | | | |
| Non-Hindu | 0.004 | 0.176 | 0.001 | -1.9 | -1.9 | 0.007 | 0.102 | 0.001 | -11.3 | -11.3 |
| **Residence** | | | | | | | | | | |
| Urban | | | | | | | | | | |
| Rural | -0.002 | -0.091 | 0.000 | -0.3 | -0.3 | 0.008 | -0.067 | -0.001 | 9.7 | 9.7 |
| **States** | | | | | | | | | | |
| Uttar Pradesh | | | | | | | | | | |
| Bihar | -0.010 | -0.174 | 0.002 | -4.3 | -4.3 | -0.011 | -0.176 | 0.002 | -32.3 | -32.3 |
| **Calculated CI** | | | -0.037 | 100.0 | | | | -0.006 | 100.0 | |
| **Actual CI** | | | -0.146 | | | | | -0.042 | | |
| **Residual** | | | -0.109 | | | | | -0.035 | | |

CI: Concentration Index; SC/ST: Scheduled Caste/Scheduled Tribe; %: Percentage.

area between line of equality and curve higher the inequality. Uttar Pradesh and Bihar witnessed a CI value of -0.15 for adolescent boys and -0.04 for adolescent girls which depicts pro-rich bias of thinness among adolescents (Fig 2A). Moreover, highest inequality of stunting was witnessed among boys (-0.16) than girls (-0.09) adolescent (Fig 2B). Additional, for the co-existence of both thinness and stunting, the inequality was increased drastically among boys and girls adolescent. For instance, the highest inequality of co-existence of both thinness and stunting was observed among boys (-0.25) compared to girls (-0.17) adolescent (Fig 2C). In

**Table 6. Estimates of decomposition analysis for contribution of various explanatory variables for stunting among adolescents aged 10–19 years.**

| Background characteristics | Adolescent boys | | | | | Adolescent girls | | | | |
|---|---|---|---|---|---|---|---|---|---|---|
| | Elasticity | CI | Absolute contribution | % contribution | | Elasticity | CI | Absolute contribution | % contribution | |
| **Age (years)** | | | | | | | | | | |
| Early adolescents (10–14) | | | | | | | | | | |
| Late adolescents (15–19) | 0.055 | 0.088 | 0.005 | -12.5 | -12.5 | 0.103 | 0.016 | 0.002 | -4.9 | -4.9 |
| **Educational status (years)** | | | | | | | | | | |
| No schooling | | | | | | | | | | |
| 1–7 | 0.028 | -0.091 | -0.003 | 6.8 | | -0.011 | -0.092 | 0.001 | -3.0 | |
| 8–9 | -0.010 | 0.064 | -0.001 | 1.6 | | -0.023 | 0.036 | -0.001 | 2.5 | |
| 10 and above | -0.022 | 0.273 | -0.006 | 15.3 | 23.6 | -0.029 | 0.272 | -0.008 | 24.0 | 23.5 |
| **Working status** | | | | | | | | | | |
| No | | | | | | | | | | |
| Yes | -0.004 | -0.156 | 0.001 | -1.6 | -1.6 | -0.002 | -0.260 | 0.001 | -1.8 | -1.8 |
| **Media exposure** | | | | | | | | | | |
| No exposure | | | | | | | | | | |
| Rare | 0.009 | -0.302 | -0.003 | 7.3 | | 0.011 | -0.221 | -0.002 | 7.3 | |
| Frequent | 0.012 | 0.120 | 0.001 | -3.6 | 3.6 | 0.024 | 0.267 | 0.006 | -19.1 | -11.8 |
| **Wealth Index** | | | | | | | | | | |
| Poorest | | | | | | | | | | |
| Poorer | -0.009 | -0.524 | 0.005 | -12.2 | | 0.001 | -0.521 | 0.000 | 1.4 | |
| Middle | -0.010 | -0.090 | 0.001 | -2.3 | | -0.005 | -0.110 | 0.001 | -1.7 | |
| Richer | -0.022 | 0.346 | -0.008 | 20.0 | | -0.015 | 0.364 | -0.005 | 16.4 | |
| Richest | -0.040 | 0.778 | -0.031 | 79.7 | 85.2 | -0.028 | 0.806 | -0.023 | 67.9 | 84.1 |
| **Caste** | | | | | | | | | | |
| SC/ST | | | | | | | | | | |
| Non-SC/ST | -0.044 | 0.089 | -0.004 | 10.1 | 10.1 | -0.033 | 0.086 | -0.003 | 8.5 | 8.5 |
| **Religion** | | | | | | | | | | |
| Hindu | | | | | | | | | | |
| Non-Hindu | -0.003 | 0.177 | -0.001 | 1.3 | 1.3 | -0.006 | 0.110 | -0.001 | 2.1 | 2.1 |
| **Residence** | | | | | | | | | | |
| Urban | | | | | | | | | | |
| Rural | 0.008 | -0.090 | -0.001 | 1.8 | 1.8 | -0.014 | -0.067 | 0.001 | -2.8 | -2.8 |
| **States** | | | | | | | | | | |
| Uttar Pradesh | | | | | | | | | | |
| Bihar | -0.026 | -0.178 | 0.005 | -11.7 | -11.7 | 0.006 | -0.182 | -0.001 | 3.1 | 3.1 |
| **Calculated CI** | | | -0.039 | 100.0 | | | | -0.033 | 100.0 | |
| **Actual CI** | | | -0.161 | | | | | -0.090 | | |
| **Residual** | | | -0.122 | | | | | -0.057 | | |

CI: Concentration Index; SC/ST: Scheduled Caste/Scheduled Tribe; %: Percentage.

other words, thinness, stunting and co-existence of both (thinness and stunting) was concentrated among poor adolescents only and it was highest among adolescent boys.

Tables 5–7 represent estimates of decomposition analysis for the contribution of various explanatory variables for thinness, stunting, and co-existence of thinness and stunting among adolescents. The percentage contribution is the column to be interpreted for depicting the percentage contribution of factors for explaining socio-economic status (SES) related inequality thinness, stunting, and co-existence of thinness and stunting among adolescents. The negative

**Table 7. Estimates of decomposition analysis for contribution of various explanatory variables for co-existence of thinness and stunting among adolescents aged 10–19 years.**

| Background characteristics | Adolescent boys | | | | | Adolescent girls | | | | |
|---|---|---|---|---|---|---|---|---|---|---|
| | Elasticity | CI | Absolute contribution | % contribution | | Elasticity | CI | Absolute contribution | % contribution | |
| **Age (years)** | | | | | | | | | | |
| Early adolescents (10–14) | | | | | | | | | | |
| Late adolescents (15–19) | 0.019 | 0.088 | 0.002 | -7.3 | -7.3 | -0.019 | 0.016 | 0.000 | 2.7 | 2.7 |
| **Educational status (years)** | | | | | | | | | | |
| No schooling | | | | | | | | | | |
| 1–7 | 0.036 | -0.092 | -0.003 | 14.5 | | -0.004 | -0.092 | 0.000 | -3.2 | |
| 8–9 | -0.001 | 0.064 | 0.000 | 0.2 | | -0.003 | 0.036 | 0.000 | 0.9 | |
| 10 and above | -0.009 | 0.269 | -0.002 | 11.0 | 25.7 | -0.003 | 0.272 | -0.001 | 6.5 | 4.2 |
| **Working status** | | | | | | | | | | |
| No | | | | | | | | | | |
| Yes | -0.012 | -0.154 | 0.002 | -8.4 | -8.4 | -0.001 | -0.260 | 0.000 | -2.6 | -2.6 |
| **Media exposure** | | | | | | | | | | |
| No exposure | | | | | | | | | | |
| Rare | 0.011 | -0.305 | -0.003 | 14.4 | | -0.005 | -0.221 | 0.001 | -9.3 | |
| Frequent | 0.004 | 0.120 | 0.000 | -2.0 | 12.4 | -0.012 | 0.267 | -0.003 | 28.2 | 18.9 |
| **Wealth Index** | | | | | | | | | | |
| Poorest | | | | | | | | | | |
| Poorer | -0.008 | -0.527 | 0.004 | -18.5 | | -0.004 | -0.521 | 0.002 | -21.0 | |
| Middle | -0.008 | -0.092 | 0.001 | -3.1 | | -0.001 | -0.110 | 0.000 | -1.4 | |
| Richer | -0.017 | 0.347 | -0.006 | 25.4 | | -0.009 | 0.364 | -0.003 | 31.1 | |
| Richest | -0.022 | 0.778 | -0.017 | 75.6 | 79.4 | -0.008 | 0.806 | -0.007 | 62.1 | 70.8 |
| **Caste** | | | | | | | | | | |
| SC/ST | | | | | | | | | | |
| Non-SC/ST | -0.007 | 0.088 | -0.001 | 2.8 | 2.8 | -0.016 | 0.086 | -0.001 | 12.8 | 12.8 |
| **Religion** | | | | | | | | | | |
| Hindu | | | | | | | | | | |
| Non-Hindu | -0.002 | 0.175 | 0.000 | 1.7 | 1.7 | 0.002 | 0.110 | 0.000 | -1.5 | -1.5 |
| **Residence** | | | | | | | | | | |
| Urban | | | | | | | | | | |
| Rural | 0.008 | -0.091 | -0.001 | 3.2 | 3.2 | 0.005 | -0.067 | 0.000 | 3.2 | 3.2 |
| **States** | | | | | | | | | | |
| Uttar Pradesh | | | | | | | | | | |
| Bihar | -0.012 | -0.175 | 0.002 | -9.4 | -9.4 | -0.005 | -0.182 | 0.001 | -8.4 | -8.4 |
| **Calculated CI** | | | -0.023 | 100.0 | | | | -0.011 | 100.0 | |
| **Actual CI** | | | -0.252 | | | | | -0.173 | | |
| **Residual** | | | -0.229 | | | | | | | |

CI: Concentration Index; SC/ST: Scheduled Caste/Scheduled Tribe; %: Percentage.

and positive signs are to make the total contribution as 100 and dependents upon the sign elasticity and concentration index (CI) of the table. The absolute contribution is the product of elasticity and CI. Moreover, the individual contribution is the division of the absolute contribution of the individual factors and total absolute contribution. Hence the magnitude of the percentage contribution depends on the elasticity and CI.

Wealth status contributed about 80 per cent to explain SES related inequality followed by educational status (15 per cent) and media exposure (12 per cent) for thinness among adolescent

boys. Whereas among adolescent girls, wealth status contributed about 66 per cent of SES related inequality followed by media exposure (58 per cent) and age (16 per cent) for thinness.

For stunting among adolescents boys, wealth status contributed 85 per cent of SES related inequality followed by educational status (24 per cent) and caste (10 per cent). In the case of adolescent girls, factors were the same as in adolescent boys i.e.; wealth status contributed 84 per cent followed by educational status (24 per cent) and caste (9 per cent) to explain SES related inequality for stunting.

For co-existence of thinness and stunting among adolescent boys, wealth status contributed 79 per cent to explain SES related inequality followed by educational status (26 per cent) and media exposure (12 per cent). Whereas in case of co-existence of thinness and stunting among adolescent girls, wealth status contributed 71 per cent to explain SES related inequality followed by media exposure (19 per cent) and caste (13 per cent).

## Discussion

Along with measuring prevalence and associated factors of thinness and stunting, this article also examined socio-economic inequality for thinness and stunting among adolescent boys and girls in two economically backward states of India, namely, Uttar Pradesh and Bihar. Furthermore, this study also focuses on the co-existence of both, i.e., thinness and stunting. Mondal and Sen (2010) also examined thinness and stunting among adolescent girls and boys; however, their study was limited to rural areas with minimal sample size [18]. Few other studies also examined thinness and stunting among adolescents. However, most of them were based on primary sample data, which was significantly less in number as compared to this study and did not examine the co-existence of thinness and stunting [19, 25]. Few other studies also examined thinness [26, 27] and stunting [28] among adolescents separately, however, adopting an approach of examining the co-existence of thinness and stunting among adolescent boys and girls makes this study different from previously available literature in various Indian settings.

The prevalence of thinness was higher among adolescent boys than in adolescent girls. Previously available studies also noticed a higher level of thinness among adolescent boys than in adolescent girls [18]. The prevalence of thinness was found to be 25.8 percent among adolescent boys and 13.1 percent among adolescent girls. Previous other studies noted a higher level of prevalence of thinness among girls and boys than in this study in different settings in India [19, 29]. In a multi-country analysis, Candler et al. (2017) noted a somewhat similar prevalence of thinness among adolescent Indian girls [30]. Furthermore, the prevalence of stunting was higher among adolescent girls (39.3%) than in adolescent boys (25.6%). Previous studies also noticed the same trends where a higher prevalence of stunting was noticed in adolescent girls than in adolescent boys in various other settings of India [1, 18]. Bhargava et al. (2020), in their study on 15–19 years of adolescents in India, found the prevalence of stunting to be 32.2 percent among boys and 34.4 percent among girls [1]. The co-existence of thinness and stunting is another prominent issue, and the study noted a higher prevalence of co-existence of thinness and stunting among adolescent boys than in adolescent girls. The unavailability of literature related to the co-existence of thinness and stunting limits our understanding of the issue in various Indian settings.

Stunting was higher among late adolescents (15–19 years) than early adolescents (10–14 years) and was more severe among late adolescent girls than their counterparts. The above finding is concordant with the previously available literature [18, 31]. However, studies in other settings found different results [12]. Stunting was severe among girls and could be attributed to their lower nutritional status than boys [32, 33]. Results significantly noted that the

odds of thinness among late adolescent girls were lower than the odds of thinness among early adolescent girls, and the same has been corroborated with the findings of Gebregiorgis, Tadesse, & Atenafu [12]. Furthermore, a study in rural Indian setting also noted the same result [34]. The thinness among early adolescents generally found to be higher than in late adolescents, possible due to increased growth spurt in the early adolescent stage as compared to the late adolescent stage [12]. Baliga, Naik, & Mallapur (2014), in their community-based rural study in Belgaum, Karnataka, noted that calorie intake deficiency was higher among early adolescent girls as compared to late-adolescent girls, which could be another possible reason of higher thinness among early adolescent girls than their counterparts [34].

Stunting and thinness among adolescents were significantly associated with various socio-economic factors. Stunting and thinness indicate the long-term cumulative inadequacies of health care services and lack of access, and an insufficient intake of food and nutrients during the early stage of childhood [18]. Several studies reported a significant association between environment, sanitation, and household pattern with undernutrition [35]. However, in this study, we could not examine any of the factors mentioned above. Education among adolescents is important factor in determining stunting and thinness among them. Results noticed that thinness and stunting among adolescent boys and girls were lower among those who had ten and above years of education than their counterparts. Previously available literature also noticed the protective effect of education on the occurrence of stunting and thinness among adolescents [36–38]. Deshmukh et al. (2006), in their study conducted in the rural area of Wardha district of Maharashtra, also highlighted a higher prevalence of stunting and thinness among those adolescents who were less educated than their counterparts [26]. Adolescents who attain higher education may receive awareness about nutrition from academic courses that may further improve their nutritional status for them [39].

The working status of adolescent boys was also found to be significantly affecting thinness among them. Results found that working status improves the odds of being stunted among adolescent boys, and the odds of co-existence of stunting and thinness were also low among working adolescent boys. Working status among adolescents may ensure income, which can further be used to improve nutritional intake. Adolescents from the richest wealth quintile were found to be having lower odds of stunting and thinness. Wealth index has also been noticed as one of the most significant contributors to the socio-economic inequality in the prevalence of stunting, thinness, and co-existence of both among adolescents. Moreover, results from the concentration curve also confirmed that stunting and thinness are concentrated among adolescents in poor households. The previously available literature is in line with this study in finding the protective feature of increasing household's wealth index or household's monthly income on stunting and thinness among adolescents [11, 31, 40]. Previous studies have linked higher household wealth to quality food and better utilization of health-care services that may be attributed to the lower odds of stunting and thinness among adolescents [41–43]. A study in North Bihar also outlined the importance of increasing wealth in reducing stunting and thinness [44]. Bhargava et al. (2020), in their study, also noted that socio-economic factors such as wealth quintiles have long term affect on undernutrition among adolescents [1].

Both thinness and stunting are the indicators of undernutrition measured through height [1]. The cut-offs used for the high prevalence of undernutrition are thinness>20% and stunting>30% [45]. Stunting and thinness among adolescents have received less attention as a public health problem in India [1]. Poor growth and height are closely associated with poverty and deprivation, reflected in Tanner's phrase,'grow this a mirror of the conditions of society [46]. Both Uttar Pradesh and Bihar are socio-economic backward states, and poor child undernutrition can be attributed to prevailing poverty in these states [47]. In our study, adolescent boys fall in the category of a high prevalence of thinness, and adolescent girls fall in the high

prevalence of stunting category. Therefore, interventions shall target to reduce thinness among adolescent boys and stunting among adolescent girls and target co-existence of stunting and thinness.

### Strengths and limitations

Due to the nature of the study design, it was not possible to establish a cause-effect relationship. The findings from this study may not be generalizable in other Indian settings because data were available for only two of the total 36 states and union territories of India. Despite the above limitations, this study has certain strengths too. The study involved a large sample size, and information on stunting and thinness were collected using standard tools. This study has examined the co-existence of stunting and thinness among adolescents and paved the way for future research that may aim to strengthen the findings.

### Conclusion

The findings from this study highlighted several important issues related to adolescent's stunting and thinness. The prevalence of thinness was higher among adolescent boys, and that of stunting was higher among adolescent girls. Furthermore, the co-existence of both stunting and thinness was higher among adolescent boys than in adolescent girls. Results also noticed that pro-poor inequality in stunting, thinness, and co-existence of both was higher for adolescent boys than in adolescent girls; it means stunting, thinness, and co-existence of both was more among adolescent boys belonging to poor households. Results from decomposition analysis revealed that the wealth index of the households explained most of the inequality in stunting, thinness, and co-existence of both among adolescents. Results further highlighted that stunting, thinness, and co-existence of both was higher among uneducated adolescents, adolescents belonging to poor households, and adolescents from SC/ST households.

At first, further research is needed to confirm factors associated with the co-existence of stunting and thinness among adolescents. The findings of this study has various policy implications. These results provide an understanding that stunting and thinness is a significant public health concern among adolescents, and there is a need to tackle the issue comprehensively. By tackling the issue comprehensively, we mean that the state government of Uttar Pradesh and Bihar shall screen, assess, and monitor the nutritional status of adolescent boys and girls. The study recommends a robust and effective implementation of nutrition education programs as a measure to prevent stunting and thinness among adolescents. Education is an important factor, and ensuring education without dropout at all levels may be a useful step in reducing stunting and thinness among adolescents. Isolated focus on either adolescent girls or boys may not be sufficient, and it is recommended that interventions shall focus towards both boys as well as girl adolescents to address different nutritional problems simultaneously. It is important to formulate nutrition interventions keeping in mind the inclusion of poorer families, as prevalence of stunting and thinness is higher among adolescents from these households.

### Author Contributions

**Conceptualization:** Pradeep Kumar, Shobhit Srivastava.

**Data curation:** Pradeep Kumar, Shobhit Srivastava.

**Formal analysis:** Pradeep Kumar, Shobhit Srivastava.

**Investigation:** Pradeep Kumar, Shobhit Srivastava.

**Methodology:** Pradeep Kumar, Shobhit Srivastava.

**Software:** Pradeep Kumar, Shobhit Srivastava.

**Supervision:** Preeti Dhillon.

**Validation:** Pradeep Kumar, Shobhit Srivastava, Preeti Dhillon.

**Visualization:** Pradeep Kumar, Shobhit Srivastava.

**Writing – original draft:** Shekhar Chauhan, Ratna Patel, Strong P. Marbaniang.

**Writing – review & editing:** Shekhar Chauhan, Ratna Patel, Preeti Dhillon.

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
