## [Decision Letter · Decision Letter 0]

23 Nov 2020

PONE-D-20-33031

Associated factors and socio-economic inequality in the prevalence of thinness and stunting among adolescent boys and girls in Uttar Pradesh and Bihar, India

PLOS ONE

Dear Dr. Marbaniang,

We look forward to receiving your revised manuscript.

Kind regards,

Vijayaprasad Gopichandran

Academic Editor

PLOS ONE

Journal Requirements:

2. As part of your revision, please complete and submit a copy of the STROBE checklist, a document that aims to improve reporting and reproducibility of observational studies for purposes of post-publication data analysis and reproducibility: (http://www.strobe-statement.org). Please include your completed checklist as a Supporting Information file. Note that if your paper is accepted for publication, this checklist will be published as part of your article.

3. In statistical methods, please clarify whether you corrected for multiple comparisons.

4. In your statistical analyses, please state whether you accounted for clustering by state/ region. For example, did you consider using multilevel models?

Reviewers' comments:

Reviewer's Responses to Questions

**Comments to the Author**

1. Is the manuscript technically sound, and do the data support the conclusions?

Reviewer #1: Yes

Reviewer #2: Yes

Reviewer #3: Yes

2. Has the statistical analysis been performed appropriately and rigorously? 

Reviewer #1: Yes

Reviewer #2: I Don't Know

Reviewer #3: Yes

3. Have the authors made all data underlying the findings in their manuscript fully available?

Reviewer #1: Yes

Reviewer #2: No

Reviewer #3: Yes

4. Is the manuscript presented in an intelligible fashion and written in standard English?

Reviewer #1: Yes

Reviewer #2: No

Reviewer #3: Yes

5. Review Comments to the Author

Reviewer #1: The reviewer must acknowledge the effort of the authors made by writing this article. Over all written and explained nicely.

• Abstract: Structurally written.

• Introduction: Looks okay

• Methods:

The basis of sample size 20594 need to be clarified a bit. Also, how the sub samples from each stratum was selected that need to be highlighted

Was there any inclusion or exclusion criteria at the time of selection of those subjects?

Though the following things mention elsewhere by the authors but still a little description of the following 3 things will help to understand better

a) Multi stage systematic sampling design need to be illustrated in a figure

b) Survey tools also need to be mentioned

c) A little description of actual data collection process also requires

• Results:

Why girls are more vulnerable to chronic malnutrition while boys are acutely malnourished? Though more boys suffer from both malnutrition?

How do you explain that total percentage of thinness and stunting among boys is little lower than total percentage of thinness and stunting among girls; but still co-existence of both is higher among boys?

Reviewer #2: This is a valuable study in the adolescent age-groups (10-19 years) from two large states of North India with good sample size and reports thinness, stunting and co-existence of both. Moreover it notes an important association of poverty with undernutrition.

Authors may consider following suggestions for revising the manuscript:

Title:

Since socio-economic factors are also associated factors, it will be better to modify the title. This is just a suggestion. A possible modification could be:

Socio-economic and demographic factors associated with prevalence of thinness and stunting among ……………

Abstract:

The results section needs to have salient findings in terms of ORs for important associations. To comply with the word limit, authors may have to modify the introduction, methodology and conclusion section. Presently the word count is 350 words which is adequate for conveying important results of the study.

Main Article:

Introduction:

As a general rule we have to support statements that mention national level facts and figures with source documents and not with smaller studies mentioning these facts and figures from somewhere else.

For example in the 6th line of the introduction, the present reference is that of a small west Bengal study which sources the information on absolute number of the adolescents in India from another resource: the SRS bulleting (India Office of the Registrar General and Census Commissioner. Census of India. SRS bulletins for years 2008, 2009, 2010, and 2011 2014. New Delhi, India: Office of the Registrar General & Census Commissioner. Available at: http://www.censusindia.gov.in/vital_statistics/SRS_Bulletins/Bulletins.html).

Similarly the 4th reference is of a study on Somalian refugees. This study mentions increased requirements of macro and micronutrients and sources the information from an authoritative reference of textbook of adolescents. You are free to use any other source document, provided it supports the increased requirement and its basis. Even the 6th reference of your manuscript is relevant.

In the second paragraph of the introduction, for the second statement: reference 7 suffices to say that stunting indicates chronic nutrition and thinness indicates acute nutrition. The reference 8 and 9 are mere small original studies that have used these indicators and do not need to be there, or are rather inappropriately there. WHO reference is enough for the use of any nutrition indicator.

As a general rule, we do not mention too many small/regional studies in the introduction. An important reference here that authors should consider introducing is the CNNS: the Comprehensive National Nutritional Survey 2016-2018 as it covers the age-groups of your interest. It also indicates the gaps in availability of nutritional indicators in all age-groups and includes information on Uttar Pradesh and Bihar.

About the sentence on undernourished adolescent girls-pregnancy-low birth weight, the reference used presently does not support this intergenerational nature of the problem as this study did not enrol any pregnant mothers. Consider replacing it with source document used by the authors in this study; please refer to the comments in the PDF document.

Rearrange the last paragraph of the introduction.

Methods:

The opening statement requires rephrasing for better readability

The first sentence of the second paragraph in the methods also requires reframing: “Due to the secondary nature of the data, authors did not approach the institutional review board for a fresh ethics approval”. This should also be the last line of the paragraph instead of the first.

It is better to write ‘written informed consent was elicited’ rather than ‘taken’ in general and ethically speaking.

Consider the suggested change in the 7th point.

Results section:

The findings of the study are very important and are available as continuous variables in terms of z-scores. Valuable information that authors can consider adding in the results (if not in the table) is the mean z-scores.

The reference number 19 needs to have a link as this is a report as there are multiple versions available. Moreover, mean z-scores are not available in the original document available from population council.

In all large scale surveys, it is not possible to do anthropometry in each and every participant. It will be important to indicate as to how many adolescents underwent height and weight measurement.

There is a repetitive use of the terms ‘likely’ and ‘likelihood’ in the section describing the table 3. Suggested rephrasing for example:

- Stunting was 79% more [OR: 1.79; CI: 1.39,2.3] in late adolescents than the late ones

- In case of adolescent girls, thinness was 52% less [OR: 0.48; CI: 0.37,0.62]; stunting was 2.25 times more [OR:2.25; CI:1.9,2.67] and presence of both 38% less [OR:0.62; CI: 0.43,0.89] in late adolescents as compared to the early ones.

Throughout the manuscript, it is preferable to use the term ‘per cent’ or ‘percent’ or ‘%’ uniformly.

Discussion: Discussion section needs significant work.

General suggestions:

- Opening paragraph should refrain from making comparisons with other studies and stick to the key summary findings of the study

- In the following sections have second level headings for each point discussed

- Separate section on strengths and limitations

Specific suggestions:

- Co-existence of thinness and stunting has clinical and public health significance and this needs to discussed with appropriate references

- Elaborate further on the socio-economic association

- Authors have done an elaborate decomposition analysis. It will be a good idea to convey this in simple terms for clinicians, public health persons and policymakers

- Discuss briefly what is the meaning of pro-poor and pro-rich inequality and its examples for understanding of general readers

- Do not restrict yourself to comparing and contrasting the findings throughout the discussion, but contextualize them between states, with other states and the upstream and downstream factors in a paragraph.

Conclusion

Needs some refinement:

- key findings

- key associations

- important implications and

- the way forward/recommendations.

The last sentence is an abrupt one which says stakeholders should increase family wealth status and reference that supports this statement also sys this in a single line. Since this is a completely new concept that ‘stakeholders should increase the wealth status’, and it is a very significant recommendation, it is either introduced somewhere in the discussion to be included in the conclusion or may be omitted.

Reviewer #3: 1Key words: Risk factors of undernutrition ,Gender inequality may be added

2.Introduction:

a)Adolescents eating habits etc may be deleted since that has not been used as an explanatory variable in this study

b)Metabolic Disorder is the only after effect of LBW that has been mentioned. There are many more important impacts of LBW

3. Methods:

a) Why is the effective sample size more than the required sample size.

b)No mention of how the sample was acquired has been done.

c)Regarding explanatory variables why education has been stratified as 1-7,8-9 and 10 above years of education

d)How has the stratification of Wealth Tax been done this may be explained

Result:

a)Fig 1 may be shown as a composite bar showing 3 bars for each condition(instead of 2) that is boys, girls, and both. Throughout boys and girls data have been dispersed extensively as a result of which no where in the entire write up there is any data describing the status of nutrition as a whole among all the adolescents irrespective of age and sex.

b)Gender inequality has not been taken care of It could have been established with proper statistical significance tests. Simply stating more or less among girls and boys is not enough.

c)Use of Concentration Index is quite superfluous for explaining the impact of income on undernutrition. Explaining with Wealth index and Decomposition analysis is more than enough.

d) In logistic Regression tables the refferents may be reversed that is Ref for Education may be "10 and above", for Working status it may be 'Yes" Wealth Index it should be "Richest" and for caste referrent may be "Richest'

e)Multivariable Logistic regression would have been very much welcome in order to make the results more robust and pinpoint the predictors

f) There are few grammatical mistakes in the inference of the tables.

Discussion:

a) Too much repetition. Make it brief and more focused esp the 1st paragraph

b)Why has the education of parents been commented on? Nowhere in the study parents' education level has been explored

c)More Studies from Bihar and UP should have been used for comparison with the data of the present study with less comparison with national level studies

6. PLOS authors have the option to publish the peer review history of their article (what does this mean?). If published, this will include your full peer review and any attached files.

Reviewer #1: No

Reviewer #2: **Yes: **Madhavi Bhargava

Reviewer #3: **Yes: **Dr Aparajita Dasgupta

---

## [Author Response · Author response to Decision Letter 0]

7 Feb 2021

Editor’s comments to author:

Response: Authors made sure that the revision followed all the guidelines laid down by the journal for submission.

2. As part of your revision, please complete and submit a copy of the STROBE checklist, a document that aims to improve reporting and reproducibility of observational studies for purposes of post-publication data analysis and reproducibility: (http://www.strobe-statement.org). Please include your completed checklist as a Supporting Information file. Note that if your paper is accepted for publication, this checklist will be published as part of your article.

Response: N/A

3. In statistical methods, please clarify whether you corrected for multiple comparisons.

Response: Changes incorporated. 

4. In your statistical analyses, please state whether you accounted for clustering by state/ region. For example, did you consider using multilevel models?

Response: The objective of the present paper was to examine the socio-economic inequality using wagstaff decomposition technique. Therefore, multilevel level modes were not used to account or to observe state/region level clustering. Moreover, the data was based on survey collected from two stated only and data was not available regional or district level therefore, accounting for clustering by state/ region was not possible. 

Comments to the Author

Reviewer 1: 

The reviewer must acknowledge the effort of the authors made by writing this article.

1. Over all written and explained nicely.

Response: Authors are thankful to reviewer for praising the overall quality of the paper.

2. Abstract: Structurally written.

Response: Thank you for acknowledging the abstract.

3. Introduction: Looks okay

Response: Thank you for reading the introduction critically and finding it okay.

4. Methods: The basis of sample size 20594 need to be clarified a bit. Also, how the sub samples from each stratum was selected that need to be highlighted

Was there any inclusion or exclusion criteria at the time of selection of those subjects?

Though the following things mention elsewhere by the authors but still a little description of the following 3 things will help to understand better.

Response: About 7932 adolescents underwent height and weight measurement. About 7539 adolescents were measure for BMI-for-age Z-score and 7586 Height -for-age Z-score. For anthropometric measure there was exclusion criteria, therefore, only selected respondents were measured for the same.

a) Multi stage systematic sampling design need to be illustrated in a figure.

Response: A written Multi stage systematic sampling design is now available in method section. 

b) Survey tools also need to be mentioned.

Response: changes incorporated. 

c) A little description of actual data collection process also requires.

Response: changes incorporated. 

5. Results:

a. Why girls are more vulnerable to chronic malnutrition while boys are acutely malnourished? Though more boys suffer from both malnutrition?

Response: We think reviewer made a point here however, he/she failed to a difference. Reviewer notices that boys suffer from both malnutrition but that is not true i.e., more girls are stunted and thin than boys. 

b. How do you explain that total percentage of thinness and stunting among boys is little lower than total percentage of thinness and stunting among girls; but still co-existence of both is higher among boys?

Response: I think reviewer failed to notice the result properly. The prevalence of thinness was much higher among boys than in girls (25.8 in boys as compared to 13.1 in girls). It is because of this difference, the co-existence of both thinness and stunting was higher among boys. The results are clearly visible in figure 1.

Reviewer 2: 

This is a valuable study in the adolescent age-groups (10-19 years) from two large states of North India with good sample size and reports thinness, stunting and co-existence of both. Moreover it notes an important association of poverty with undernutrition.

Authors may consider following suggestions for revising the manuscript:

1. Title:

A. Since socio-economic factors are also associated factors, it will be better to modify the title. This is just a suggestion. A possible modification could be: Socio-economic and demographic factors associated with prevalence of thinness and stunting among

Response: The authors feel that there is no need to change the title as the current title is justifiable. The main emphasis is one socio-economic inequality and that is depicted through the title.

2. Abstract:

A. The results section needs to have salient findings in terms of ORs for important associations. To comply with the word limit, authors may have to modify the introduction, methodology and conclusion section. Presently the word count is 350 words which is adequate for conveying important results of the study.

Response: comment incorporated. 

3. Main Article:

Introduction:

Query: As a general rule we have to support statements that mention national level facts and figures with source documents and not with smaller studies mentioning these facts and figures from somewhere else.

For example in the 6th line of the introduction, the present reference is that of a small west Bengal study which sources the information on absolute number of the adolescents in India from another resource: the SRS bulleting (India Office of the Registrar General and Census Commissioner. Census of India. SRS bulletins for years 2008, 2009, 2010, and 2011 2014. New Delhi, India: Office of the Registrar General & Census Commissioner. Available at: http://www.censusindia.gov.in/vital_statistics/SRS_Bulletins/Bulletins.html).

Response: Thanks for the suggestion. Now the source of information has been cited from the original source.

Query: Similarly the 4th reference is of a study on Somalian refugees. This study mentions increased requirements of macro and micronutrients and sources the information from an authoritative reference of textbook of adolescents. You are free to use any other source document, provided it supports the increased requirement and its basis. Even the 6th reference of your manuscript is relevant.

Response: Now the reference has been taken care.

Query: In the second paragraph of the introduction, for the second statement: reference 7 suffices to say that stunting indicates chronic nutrition and thinness indicates acute nutrition. The reference 8 and 9 are mere small original studies that have used these indicators and do not need to be there, or are rather inappropriately there. WHO reference is enough for the use of any nutrition indicator.

Response: Now the reference has been corrected as per the suggestion.

Query: As a general rule, we do not mention too many small/regional studies in the introduction. An important reference here that authors should consider introducing is the CNNS: the Comprehensive National Nutritional Survey 2016-2018 as it covers the age-groups of your interest. It also indicates the gaps in availability of nutritional indicators in all age-groups and includes information on Uttar Pradesh and Bihar. About the sentence on undernourished adolescent girls-pregnancy-low birth weight, the reference used presently does not support this intergenerational nature of the problem as this study did not enroll any pregnant mothers. Consider replacing it with source document used by the authors in this study; please refer to the comments in the PDF documents. Rearrange the last paragraph of the introduction.

Response: Thanks for the comments. Now the reference has been replace with the reference from the original source. Also, the last paragraph has been rearranged. 

Methods:

Query: The opening statement requires rephrasing for better readability

The first sentence of the second paragraph in the methods also requires reframing: “Due to the secondary nature of the data, authors did not approach the institutional review board for a fresh ethics approval”. This should also be the last line of the paragraph instead of the first.

It is better to write ‘written informed consent was elicited’ rather than ‘taken’ in general and ethically speaking. Consider the suggested change in the 7th point.

Response: comment incorporated.

Results section:

Query: The findings of the study are very important and are available as continuous variables in terms of z-scores. Valuable information that authors can consider adding in the results (if not in the table) is the mean z-scores.

Response: z-scores are now provided.

Query: The reference number 19 needs to have a link as this is a report as there are multiple versions available. Moreover, mean z-scores are not available in the original document available from population council.

Response: Now we have added a separate table (table 2b) for z-score. Also, we have provided the required link.

Query: In all large scale surveys, it is not possible to do anthropometry in each and every participant. It will be important to indicate as to how many adolescents underwent height and weight measurement.

Response: 7932 adolescents underwent height and weight measurement. About 7539 adolescents were measure for BMI-for-age Z-score and 7586 Height -for-age Z-score. 

Query: There is a repetitive use of the terms ‘likely’ and ‘likelihood’ in the section describing the table 3. Suggested rephrasing for example:

- Stunting was 79% more [OR: 1.79; CI: 1.39,2.3] in late adolescents than the late ones

- In case of adolescent girls, thinness was 52% less [OR: 0.48; CI: 0.37,0.62]; stunting was 2.25 times more [OR:2.25; CI:1.9,2.67] and presence of both 38% less [OR:0.62; CI: 0.43,0.89] in late adolescents as compared to the early ones.

Response: Thanks for the suggestions. We have rephrased the description of table 3.

Query: Throughout the manuscript, it is preferable to use the term ‘per cent’ or ‘percent’ or ‘%’ uniformly.

Response: Thanks for pointing out. Now the term ‘per cent’ is used uniformly throughout the manuscript.

Discussion: 

Query: Discussion section needs significant work.

Response: The discussion section has been improved significantly as per given suggestion.

Query: General suggestions:- Opening paragraph should refrain from making comparisons with other studies and stick to the key summary findings of the study.

Response: Authors are thankful to the reviewer for raising this issue. We have made the brief changes to the paragraph. However, authors would like to reiterate that comparisons were not made for findings. Authors have cited other studies in the first paragraph of discussion to present the uniqueness of this study. By citing other studies, authors intended to depict how this study differs from previously available studies. Therefore, authors feel that these studies should not be omitted from the discussion section. Furthermore, in second paragraph of the discussion where authors made comparison with previous studies is done to depict changes over time in the prevalence of stunting and thinness. Authors strongly feel that these studies shall not be omitted from the discussion as these studies are providing depth to the discussion.

Query: In the following sections have second level headings for each point discussed

- Separate section on strengths and limitations.

Response: Authors have provided separate heading for strengths and limitations section.

Specific suggestions:

Query: Co-existence of thinness and stunting has clinical and public health significance and this needs to discussed with appropriate references

Response: Authors agreed that co-existence of thinness and stunting has clinical and public health significance and therefore as per given suggestion, we have highlighted this importance by citing some important studies in the discussion. 

Query: Elaborate further on the socio-economic association.

Response: The authors have focused on socio-economic association in the discussion section as suggested.

Query: Authors have done an elaborate decomposition analysis. It will be a good idea to convey this in simple terms for clinicians, public health persons and policymakers.

Response: The decomposition analysis is defined properly in the method section of the article.

Query: Discuss briefly what is the meaning of pro-poor and pro-rich inequality and its examples for understanding of general readers.

Response: Pro-rich and pro-poor has been defined properly in the method section under the heading ‘concentration index.’

Query: Do not restrict yourself to comparing and contrasting the findings throughout the discussion, but contextualize them between states, with other states and the upstream and downstream factors in a paragraph.

Response: The authors have improved the discussion section on the suggested lines.

Conclusion

Query: Needs some refinement:

- key findings

- key associations

- important implications and

- the way forward/recommendations.

Response: The conclusion section has also been improved as suggested by the reviewer.

Query: The last sentence is an abrupt one which says stakeholders should increase family wealth status and reference that supports this statement also sys this in a single line. Since this is a completely new concept that ‘stakeholders should increase the wealth status’, and it is a very significant recommendation, it is either introduced somewhere in the discussion to be included in the conclusion or may be omitted.

Response: As per the suggestion from the reviewer, we have omitted the last line from conclusion section stating ‘stakeholders should increase the wealth status.’

Reviewer 3: 

1. Key words: Risk factors of undernutrition. Gender inequality may be added.

Response: Changes incorporated

2.Introduction:

a)Adolescents eating habits etc may be deleted since that has not been used as an explanatory variable in this study.

Response: The authors do agree with the reviewer comment. Now we have deleted this statement regarding adolescent eating habits in the manuscript. 

b)Metabolic Disorder is the only after effect of LBW that has been mentioned. There are many more important impacts of LBW.

Response: Thank you for this important suggestion. Now we have included some more important consequence of LBW

3. Methods:

a) Why is the effective sample size more than the required sample size.

Response: About 7932 adolescents underwent height and weight measurement. About 7539 adolescents were measure for BMI-for-age Z-score and 7586 Height -for-age Z-score. 

b)No mention of how the sample was acquired has been done.

Response: Only those adolescents were added in the sample who underwent height and weight measurement.

c)Regarding explanatory variables why education has been stratified as 1-7, 8-9 and 10 above years of education.

Response: The data provides the levels as such, no other question was asked for measuring educational status. 

d)How has the stratification of Wealth Tax been done this may be explained.

Response: Changes incorporated. 

Result:

a)Fig 1 may be shown as a composite bar showing 3 bars for each condition(instead of 2) that is boys, girls, and both. 

Response: Thanks for the comments. UDAYA data provide estimates separately for boys and girls; therefore a composite estimate (prevalence) is not provided in the study.

b) Throughout boys and girls data have been dispersed extensively as a result of which no where in the entire write up there is any data describing the status of nutrition as a whole among all the adolescents irrespective of age and sex.

Response: The overall nutrition status of adolescents irrespective of age and sex is not available in the data sets. The sampling design of the survey does not permit to give overall estimates of nutrition status of adolescents. And we have writing it into the limitation section. Therefore, we have provided the estimates separately for girls and boys.

c) Gender inequality has not been taken care of. It could have been established with proper statistical significance tests. Simply stating more or less among girls and boys is not enough.

Response: The objective of the paper was to examine the socio-economic differentials. Moreover, UDAYA data provide estimates separately for boys and girls; therefore a composite estimate (prevalence) is not provided in the study.

c) Use of Concentration Index is quite superfluous for explaining the impact of income on undernutrition. Explaining with Wealth index and Decomposition analysis is more than enough.

Response: Thanks for the observation. Estimating Concentration Index and drawing a concentration curve is important as it shows that upto what extent the outcome variable was concentrated among poor or rich. Therefore, it is important to estimate Concentration Index before applying Decomposition analysis.

d) In logistic Regression tables the refferents may be reversed that is Ref for Education may be "10 and above", for Working status it may be 'Yes" Wealth Index it should be "Richest" and for caste referrent may be "Richest'

Response: Changes incorporated. 

e)Multivariable Logistic regression would have been very much welcome in order to make the results more robust and pinpoint the predictors.

Response: We have already used multivariable logistic regression. 

f) There are few grammatical mistakes in the inference of the tables.

Response: Grammatical mistakes have been corrected.

Discussion:

a) Too much repetition. Make it brief and more focused esp the 1st paragraph.

Response: The first paragraph has been modified as suggested. Furthermore, discussion has been improved significantly.

b)Why has the education of parents been commented on? Nowhere in the study parents' education level has been explored.

Response: Authors agreed with the reviewer on this point. Accordingly, we have omitted the text related to parent’s education from the discussion section.

c)More Studies from Bihar and UP should have been used for comparison with the data of the present study with less comparison with national level studies.

Response: Due to unavailability of studies from Bihar and UP, we had to make comparison with the data of present study with national level studies. However, we have modified the discussion and cited a few studies from Bihar and UP in the discussion section at relevant place.

---

## [Editor Report · Decision Letter 1]

9 Feb 2021

Associated factors and socio-economic inequality in the prevalence of thinness and stunting among adolescent boys and girls in Uttar Pradesh and Bihar, India

PONE-D-20-33031R1

Dear Dr. Marbaniang,

We’re pleased to inform you that your manuscript has been judged scientifically suitable for publication and will be formally accepted for publication once it meets all outstanding technical requirements.

Kind regards,

Vijayaprasad Gopichandran

Academic Editor

PLOS ONE
---

## [Editor Report · Acceptance letter]

15 Feb 2021

PONE-D-20-33031R1 

Associated factors and socio-economic inequality in the prevalence of thinness and stunting among adolescent boys and girls in Uttar Pradesh and Bihar, India 

Dear Dr. Marbaniang:

I'm pleased to inform you that your manuscript has been deemed suitable for publication in PLOS ONE. Congratulations! Your manuscript is now with our production department. 

Kind regards, 

on behalf of

Dr. Vijayaprasad Gopichandran 

Academic Editor

PLOS ONE